# Peptide−Calcium Chelate from Antler (*Cervus elaphus*) Bone Enhances Calcium Absorption in Intestinal Caco-2 Cells and D-gal-Induced Aging Mouse Model

**DOI:** 10.3390/nu14183738

**Published:** 2022-09-10

**Authors:** Zhaoguo Wang, Xiaorui Zhai, Jiayuan Fang, Hongyan Wu, Yunyun Cheng, Yuan Gao, Xi Chen, Shuo Zheng, Songcai Liu, Linlin Hao

**Affiliations:** 1College of Animal Science, Jilin University, Changchun 130061, China; 2College of Public Health, Jilin University, Changchun 130061, China

**Keywords:** antler bone, peptides, calcium absorption, intestinal Caco-2 cells, aging mouse model

## Abstract

Antler bone calcium (AB−Ca) and bioactive peptides (ABPs) were extracted from antler bones (*Cervus elaphus*) to maximize their value. In this study, 0.14 g calcium was obtained from 1 g antler bone. The peptide−calcium chelate rate was 53.68 ± 1.80%, and the Gly, Pro, and Glu in ABPs were identified to donate most to the increased calcium affinity through the mass spectrometry. Fourier transform infrared spectroscopy showed that calcium predominantly interacted with amino nitrogen atoms and carboxyl oxygen atoms, thereby generating a peptide–calcium chelate. The peptide−calcium chelates were characterized using scanning electron microscopy. A Caco-2 cell monolayer model showed that ABPs significantly increased calcium transport. Furthermore, the D-gal-induced aging mouse model indicated that the ABPs + AB−Ca group showed higher Ca and PINP levels, lower P, ALP, and CTX-1content in serum, and considerably higher tibia index and tibia calcium content. Results showed that ABPs + AB-Ca increased bone formation and inhibited bone resorption, thereby providing calcium supplements for ameliorating senile osteoporosis (SOP).

## 1. Introduction

Calcium deficiency is prevalent, especially in the elderly [1], with serious clinical consequences such as senile osteoporosis (SOP), which is characterized by reduced bone mass, increased bone weakness, and degraded microstructural bone structures [2,3,4]. The number of fractures caused by SOP exceeds 8.9 million per year, primarily in Europe and America [5], among which, hip fracture is the most destructive in the elderly, with a mortality rate of up to 20% one year after the fracture [4,6].

Calcium supplementation, one of the most effective approaches to reduce the risk of SOP [7], is difficult in the elderly because of the degenerative decay of intestinal absorption and renal reabsorption [8]. Research has shown that calcium absorption can be promoted by bisphosphonate (BPs), which inhibits osteoclast activity and attenuates bone loss [9]. Recombinant, human, parathyroid hormone (rhPTH) improves bone mass by boosting calcium absorption in the intestine and renal reabsorption [10]. Nevertheless, some negative effects of these treatments have been reported, including skeletal pain [11], osteosarcoma [12], hypercalcemia [11], and jaw osteonecrosis [12]. However, calcium carbonate is another highly efficient strategy for calcium supplementary products to avoid SOP. Unfortunately, calcium carbonate has limited therapeutic advantages in practice, with poor absorption and bioavailability owing to its insolubility in the intestine [13,14].

Peptides have been demonstrated to have considerable benefits in calcium absorption based on their calcium-binding activity; they are regarded as ideal candidates to assist in calcium supplementation [15,16]. Peptides from pig-bone collagen [17], chicken-foot broth byproduct [18], Alaska pollock skin [19], tilapia [20], wheat germ [21], and cucumber seeds [22] have been observed to have calcium-binding activity. The study found that peptide–calcium chelate promoted Ca absorption through the intestinal wall by sustaining the soluble Ca concentration in the intestine [23]. A low chelation rate is a major obstacle to large-scale application of commercial production; the calcium-binding activity of peptides generated from chicken metatarsus bone [18] and Alaska pollock skin [19] is 235.7 ± 20.0 mg/g and 11.52 ± 2.23 nmol/µmol, respectively. Therefore, identifying underutilized food products with a high chelation rate is important, especially for elderly individuals with calcium deficiency. Unlike other mammalian tissues or organs, the antler velvet is the only mammalian organ that undergoes periodic replacement [24]. Antler velvet is a traditional Chinese functional food that contains various biological elements and is often used as a dietary and medication supplement [24]. The antler bone was located in the basal section. Analysis of antler bone has revealed that it contains high amounts of protein [25], mineral calcium [26], and phosphorus [27]. The calcium–phosphorus ratio of antler bone is greater than 1:1 and is easily absorbed by the human body [28,29]. Antler bone extracts enhance physical strength [30], serve as antioxidants [31], anti-inflammatory [32], and anti-cancer [33], and improve sexual function [30]. China is an important deer-raising and antler-producing country [34]. Traditionally, antler bone has a low market value owing to severe ossification [28]. However, few studies have highlighted that antler bones serve as sources of calcium and peptide.

Therefore, this study aimed to investigate the effect of velvet antler peptide-calcium chelate on calcium absorption in the elderly. Calcium and peptides were obtained from antler bone and identified by atom-absorption spectroscopy (AAS) and mass spectrometry (MS). The peptide–calcium chelate was developed and identified using Fourier transform infrared spectroscopy (FTIR) and scanning electron microscopy (SEM). Finally, the calcium transport capacity of peptide–calcium chelate was investigated in vitro, and bone metabolism in an aging-mouse model was investigated in vivo. The results of this work provide fundamental theoretical research on the use of antler-bone peptides to fabricate peptide–calcium chelates, which might be employed as a functional food component to provide calcium fortification for ameliorating SOP.

## 2. Materials and Methods

### 2.1. Reagents and Materials

Fresh antler bone was provided by CHANG SHENG Deer Industry Co., Ltd. (Changchun, Jilin, China) and preserved at −20 °C until use. Malic acid was purchased from Wan Bang Industrial Co., Ltd. (Zhengzhou, China). Flavourzyme (≥20 units/mg) and d-galactose were obtained from Yuanye Biotechnology (Shanghai, China). DMEM/F-12, penicillin-streptomycin, fetal bovine serum (FBS), and other reagents were obtained from Gibco (Carlsbad, CA, USA) and Sigma-Aldrich (St. Louis, MO, USA), respectively. Other reagents were commercially available and of analytical grade or higher.

### 2.2. Bone Calcium Solution and Total Ash Determination

The preparation of antler-bone calcium was evaluated by applying the method of Weaver et al. with a few adjustments [35]. The antler bone was sliced using a bone slicer, cleaned in deionized water to neutral pH, and then dried at room temperature (RT). After drying, the bone was ground into powder. Antler bone powder (1 g) was weighed and combined with 10 mL malic acid solution, which was prepared at a concentration of 5%, then stirred at 55 °C for 70 min. The solution was centrifuged at the process of 10,000× *g* for 10 min, and RT, yielding a clear supernatant containing the bone calcium solution (AB−Ca). AB−Ca was then lyophilized and preserved at −20 °C for use as CaCl_2_ replacement. We weighed the antler bone section (2 g), placed it into a muffle furnace (SGM-M16/10, SIGMA, Shanghai, China) and dried for over 12 h at 550 °C until it reached a constant weight. The ash weight was calculated, the sample was diluted with deionized water, and calcium was tested using AAS (PE-700, Agilent, Santa Clara, CA, USA).

### 2.3. Peptide Extraction

The extraction of antler bone peptides was performed according to Wu’s method [17] with slight modifications. Antler bone slices were diced with a bone slicer, rinsed with deionized water to a neutral pH, and dried at RT. In prior study, Flavourzyme was used to obtain antler bone peptides (ABPs) with a pH of 8.5, a temperature of 50 °C, a hydrolysis period of 5 h, a substrate concentration of 10%, and an enzyme dose of 7000 U/g. After enzymatic digestion, enzyme inactivation (100 °C, 5 min), centrifugation (4 °C, 10 min, 10,733× *g*), collected the supernatant, lyophilized, and then preserved at −20 °C for backup.

### 2.4. MS Identification and Sequence Analysis of ABPs

MS was used to identify lyophilized ABPs using an Orbitrap XL instrument (Thermo Fisher Scientific, Waltham, MA, USA). (A) Formic acid (1%) and (B) CH_3_CN (1% formic acid) were used as aqueous mobile phases. At a flow rate 400 L/min for 50 min, a linear gradient of B from 3% to 44% was utilized to elute ABPs. The MS parameters were as follows: 2.2 kV spray voltage, 5 μL/min flow rate, and 350.0–1550.0 *m/z* mass range. The original peptide sequences from the deer antler (*Cervus elaphus*) database were identified using the UniProt database.

### 2.5. Calcium Binding Capacity of ABPs

Lyophilized ABPs were dissolved in deionized water and AB−Ca was added at the same mass ratio (1:1). The solution was stirred at pH 7.5, 50 °C for 60 min. After the chelation reaction, the mixture was added to absolute ethanol (*v*/*v* 1:9), centrifuged, gathered, freeze-dried, and labeled as a peptide−calcium chelate. AAS was used to assess calcium concentration in the supernatant. With minor modifications, calcium-binding capacity was calculated using the method of Wu et al. [17]. The peptide−calcium chelate was lyophilized for further use. Commercially available bone peptides from cattle were used as controls.

### 2.6. FTIR Measurement

Samples (1 mg) of ABPs or peptide–calcium chelate were crushed with 100 mg of dry KBr and then compressed into a clear piece for evaluation. An FTIR instrument (Thermo Nicolet iS5, Madison, WI, USA) was used to obtain absorption spectra in the 400–4000 cm^−1^ region. Omnic 8.2 software (Thermo Nicolet, Madison, WI, USA) was used to analyze the peak signals in the spectra.

### 2.7. SEM Measurement

The samples (ABPs and peptide−calcium chelate) were added to the plate and sprayed with gold. Images of ABPs and peptide–calcium chelates were acquired by SEM (SU8010, HITACHI, Japan) at a voltage of 5 kV.

### 2.8. The Caco-2 Cells Monolayers Establishment

Caco-2 cells were cultured at 37 °C in 5% CO_2_ in fresh medium (DMEM/F-12, 10% FBS, and 1% penicillin-streptomycin). Adherent cells were dissociated with 0.25% trypsin and then inoculated in 24-well Transwell plates with polycarbonate membranes (TCS001024, JET BIOFOL, Guangzhou, Guangdong, China) at a density of 1 × 10^5^ cells/mL. An electrical-resistance meter was used to measure the resistance values of the intestinal Caco-2 cell monolayers (Millicell-ERS, Millipore Corp., Burlington, MA, USA), and the cell culture medium was replenished on alternate days. The cell model was established when the transepithelial electrical resistance (TEER) exceeded 300 Ω/cm^2^, and calcium transport experiments were performed.

### 2.9. Caco-2 Cells Monolayer Calcium Transport Studies

Calcium transport was measured following Sun’s method [36], with slight modifications. The cell model was cultured in HBSS (without Ca and Mg) at 37 °C with 5% CO_2_ for 30 min. The apical side was then supplied with AB−Ca (equivalent to calcium 60 μg/well) and ABPs + AB−Ca (ABPs: 60 μg/well; AB−Ca: equivalent to calcium 60 μg/well). Each sample was pre-mixed with Caco-2 cells at 37 °C for 30 min. Equal amounts of CaCl_2_ were used as the control group. AAS was used to measure calcium concentration in 500 μL of HBSS from the basolateral side (BS) at given points in time (30, 60, 90, 120, 150, 180, and 240 min). The volume of the BS was maintained by promptly adding 500 μL of new HBSS. The total Ca transport to the BS was calculated according to Wu et al. [17] as follows:$$\mathrm{Bn}=\mathrm{An}+0.5\;\times \;\overset{n\;-\;1}{\underset{k=1}{\sum }}\;\mathrm{Ak}$$
where “Bn” indicates the total amount of Ca transported to the BS of each well at the given time-point (unit: g/well). “An” indicates the Ca concentration of the HBSS in the BS of each well at the given time-point (unit: g/mL); “0.5” indicates that the Ca concentration was determined by taking 0.5 mL of HBSS from the BS of each well; n (independent variable) might be 1, 2, 3, 4, 5, 6, or 7, corresponding to time points of 30, 60, 90, 120, 150, 180, and 240 min, respectively.

### 2.10. Animals

The Liaoning Changsheng Experimental Animal Center (Benxi, Liaoning, China) provided male ICR mice (eight-weeks-old). The mice had free adequate food and sterile water with maintained controlled temperature, humidity, and light (22 ± 3 °C, 60 ± 5%, and diurnal cycle of 12 h) in the environment at the Animal Center of Jilin University (Permission number: KT202003150). The experimental protocol was approved by the Institutional Animal Care and Use Committee (IACUC).

### 2.11. Feeding Procedures and Daily Observation of Physical Condition

After a week of adaptation, the mice were randomly divided into four groups: control, CaCl_2_ group, AB−Ca, and the ABPs + AB−Ca group (n = 8). D-gal was administered subcutaneously to all mice at a dose of 400 mg/kg for 58 days [37]. On day 28, the mice in the NC group were administered 0.9% normal saline orally, those in the CaCl_2_ group with CaCl_2_ (30 mg calcium per 100 g), those in the AB−Ca group with AB−Ca (30 mg calcium per 100 g), and those in the ABPs + AB−Ca group with ABPs (30 mg ABPs per 100 g) and AB−Ca (30 mg calcium per 100 g) for 30 days before sacrifice. The AIN-93M diet [38] formula and deionized water were freely available to all the mice. The body weight (BW) of each mouse was measured and recorded once per week using an electronic balance. Figure 1 depicts the schedule of the complete mouse experimental process.

### 2.12. Serum Biochemistry

The mice were sedated with isoflurane at the conclusion of the experimental feeding period and suborbital blood was obtained immediately. Blood samples were coagulated for 2 h at RT, centrifuged at RT for 20 min at 2000× *g* to extract the serum, and then preserved at −80 °C before use. Serum biochemical levels of Ca and P (Beijing Solarbio, Beijing, China), ALP (Nanjing Jiancheng, Nanjing, China), PINP (ZCI BIO, Shanghai, China), and CTX-1 (Elabscience, Wuhan, China) were measured using commercially available kits.

### 2.13. Estimation of Visceral Indices

The mice were euthanized by fast cervical dislocation on day 58. Visceral indices were calculated by weighing the heart, liver, spleen, lungs, and kidneys.
$$\mathrm{visceral}\;\mathrm{index}\;\left(100\%\right)=\frac{\mathrm{weight}\;\mathrm{of}\;\mathrm{viscera}}{\mathrm{body}\;\mathrm{weight}}$$

### 2.14. Micro-CT Analysis

All left tibias were extracted and dissected, and any tissue attached to the tibia was removed. Vernier caliper and electronic balance were used to determine the length and weight of the tibias, respectively. A dual-energy X-ray absorptiometer (AG CT50, Zurich, Switzerland) was used to analyze the tibial indices.

### 2.15. Determination of Total Ash

Bone calcium levels were analyzed using AAS, and all left tibia bones were weighed, dried, and dissolved in deionized water according to a previously described antler bone calcium determination method.

### 2.16. Statistical Analyses

All experimental data are shown as the mean ± SD of at least three independent experiments. One-way or two-way variance (ANOVA) was used to measure statistical differences among the groups. All statistical analyses were performed using GraphPad Prism 9.0 (San Diego, CA, USA). Differences were considered statistically significant at *p* < 0.05.

## 3. Results and Discussion

### 3.1. Antler Bone Calcium (AB−Ca)

Antler bone had a calcium concentration of 34.36 ± 0.23% by AAS measurement, similar to that of the elemental calcium in anhydrous calcium chloride (at 36%). The gross composition of antler bone is similar to that of other bones [27] with a high elemental calcium concentration, which agrees with the findings of Landete-Castillejos et al. [25]. After treatment with 5% malic acid at 55 ℃ for 70 min, 0.14 g calcium in the solution was obtained from 1 g antler bone. Therefore, AB−Ca provides a source of calcium for chelation reactions with peptides, as proven in this study.

### 3.2. Calcium-Binding Capacity and Amino Sequences Analysis of ABPs

In our study, ABPs were shown to have a calcium-binding capacity of 53.68 ± 1.80%, substantially higher than the standard of cattle-bone collagen peptide (36.69 ± 0.56%). The capacity of peptides to bind calcium is related to their molecular weights and amino acid compositions. Thus, the molecular weight and amino acid sequences of the ABPs were determined using mass spectrometry. The molecular weight of the ABPs ranged from 567.27 Da to 1649.76 Da, as shown in Table 1. These findings are supported by prior research that revealed that lower molecular mass peptides have a proclivity for combining with calcium ions [39,40]. The amino acid sequences of the ABPs corresponded well with the UniProt database (Table 1). The results showed that the sequences Gly-Pro-Gly, Pro-Gly-Pro, Gly-Ala-Pro, and Gly-Glu-Pro-Gly were highly repeated in ABPs, which have been proven to be associated with bone remodeling in previous studies [41]; Gly, Pro, Glu, and Ala appeared in almost every peptide. Gly, Pro, and Glu are abundant in Pacific cod-bone peptides, which are essential for calcium-binding activities [42]. The high amounts of Gly and Glu were linked to high calcium affinity in Hoki-bone hydrolysates, indicating that the peptide sequences in our analysis were comparable [43]. The high chelation rate of peptide–calcium chelate obtained by ABPs due to its Gly, Pro, and Glu abundance is effective for some difficult calcium supplementation targets, such as SOP.

### 3.3. FTIR Measurement of Peptide–Calcium Chelate

FTIR spectroscopy has been used to reveal the formation of organic functional groups of peptides and calcium ions [44], similar to that of an excellent instrument for understanding the structures of peptides [45]. Figure 2 illustrates the spectra of the ABPs and peptide–calcium chelates. After the chelation reaction, the high absorption peak at 3442.31 cm^−1^ shifted to 3422.31 cm^−1^, illustrating that N−H participated in the formation of hydrogen bonds after chelation with AB−Ca [46]. The absorption of the amide-I vibration at 1641.13 cm^−1^ indicates that the stretching vibration of C=O was displaced to 1633.41 cm^−1^, following chelation with AB−Ca. The stretching of −COOH coupled with AB−Ca to generate −COO−Ca causes the peak at 1066.92 cm^−1^ to shift to a higher valley peak at 1125.26 cm^−1^. The wavenumber shifted from 515.38 cm^−1^ to 526.47 cm^−1^ as the C−H and N−H bonds were replaced by N−Ca bonds [47]. In summary, ABPs exploit carboxyl oxygen and amino nitrogen to chelate AB−Ca to synthesize peptide–calcium chelates. The results of the cucumber seed peptide–calcium chelate were comparable to those of this study [22].

### 3.4. SEM Measurement of Peptide–Calcium Chelate

The microstructures of ABPs and peptide–calcium chelates are shown in Figure 3. The surface structure of the ABPs was smooth with a floccule structure or granular substance attached to the surface, which may have been caused by the adsorption of a small amount of free calcium in the peptide extraction process. After the chelation reaction, the peptide–calcium chelate displayed a denser cluster structure, and many spherical aggregates were observed under magnification, indicating that ABPs chelated with AB−Ca to form denser particles [48]. The differences in the microstructures shown in Figure 3 might be due to the interaction between ABPs and peptide–calcium chelate, forming a new structure. The study of desalinated, duck egg-white peptides showed consistent results, and the chelation of peptides and calcium changed the morphological characteristics [49].

### 3.5. ABPs Significantly Improved Calcium Transport in Intestinal Caco-2 Cells

The Caco-2 cell monolayer model has been effectively used to simulate absorption, including mineral elements [50] and peptides [51]. Caco-2 cells (human intestinal adenocarcinoma cells) are suitable models for simulating human intestinal cells in vitro [16]. In this study, Caco-2 cells naturally differentiated into a monolayer cell model during 19 days of cell culture [52], and the TEER values of the monolayer were measured (Figure 4A).

Subsequently, ABPs’ calcium transport activity was assessed. Our results showed that ABPs significantly improved calcium transport activity compared with the NC and AB−Ca groups in the Caco-2 monolayer model (Figure 4B). Within 240 min, calcium absorption increased with time. In the first 30 min, the calcium transport rate of the ABPs + AB−Ca group was especially notable; subsequently, calcium transport was sluggish. The rates of calcium transport in the ABPs + AB−Ca group were 3.88-, 2.13-, 2.55-, 3.13-, 2.38-, 2.31-, and 2.30-fold greater than those in the NC group, and 1.28-, 1.24-, 1.13-, 1.15-, 1.20-, 1.19-, and 1.19-fold greater than those of the AB−Ca group at 30, 60, 90, 120, 150, 180, and 240 min, respectively. Moreover, the AB−Ca group showed improved calcium transport by 3.03-, 1.71-, 2.26-, 2.74-, 1.98-, 1.94, and 1.93 times over that of the NC group (from 30 to 240 min). Organic calcium has a higher solubility and absorption rate and causes less irritation to the gastrointestinal tract than inorganic calcium [53]. In vitro, calcium malate from eggshells may improve calcium absorption, which is comparable to our findings.

Owing to the degenerative decline in intestinal calcium absorption caused by SOP, peptide–calcium chelates with a high chelation rate bonded calcium for transport, easily released ions, and improved intestinal calcium absorption [54]. Transcellular and paracellular pathways are the two primary mechanisms of intestinal calcium absorption. Previous research has suggested that peptides from casein [16] acted as calcium carriers by interacting with TRPV6 and play a role in the Ca^2+^ transcellular pathway. According to the results of desalted duck egg-white peptides, the paracellular route has a minor effect on calcium transport [55]. Pig-bone collagen peptides improved calcium transport in intestinal Caco-2 cells, which reversed the inhibitory effects of phosphate and phytate [17]. Small peptides [56] have been demonstrated to be absorbed directly in the intestine. In vitro experiments revealed that ABPs isolated from antler bones greatly increased calcium uptake. Whether ABPs affect TRPV6 via the Ca^2+^ transcellular pathway participating in the promotion of calcium transport requires further investigation.

### 3.6. No Substantial Variation in Weight Gain and Visceral Index Were Found among the Groups

SOPs are major public health issues [57], and their occurrence and severity increase dramatically as the population ages. Aging is generally associated with a decline in bone mass. Nevertheless, previous research has universally accepted the D-gal-induced aging mouse model (Appendix A [37,58,59,60,61,62,63,64]), resulting in bone loss [37]. We used a D-gal-induced (400 mg/kg/day) aging model to identify the unexpected role of peptide–calcium chelate in bone metabolism (Appendix A).

As shown in Figure 5A, no significant variation in weight gain was found among the groups (*p* > 0.05). Visceral indices showed no statistical differences among the groups, as shown in Figure 5B (*p* > 0.05). These data indicate that the current dose of CaCl_2_, AB−Ca, and ABPs + AB−Ca groups had no evident effect on normal growth and health. No diarrhea was observed during the experiment. This result was similar to those of studies conducted in rodents [65,66], indicating that antler polypeptides are harmless, without teratogenic or hepatoprotective effects.

### 3.7. ABPs + AB−Ca Group Had Significant Increase in Ca and PINP and Decrease in P, ALP, and CTX-1 Content in the Serum

Table 2 shows that the ABPs + AB−Ca group had higher serum Ca concentrations than the NC, CaCl_2_, and AB−Ca groups (*p* < 0.05). The ABPs + AB−Ca group had substantially lower serum P levels than the other groups (*p* < 0.05). The AB−Ca and ABPs + AB−Ca groups had significantly lower serum ALP and CTX-1 levels than the NC and CaCl2 groups (*p* < 0.05), and the AB−Ca and ABPs + AB−Ca groups showed no significant difference (*p* > 0.05). The results revealed that ABPs + AB−Ca raised serum Ca levels, similar to the results of peptides from Pacific cod bone, due to the lower fecal and urine calcium excretion [67]. P is an important inorganic nutrient in bone metabolism [68], and increased bone resorption leads to increased serum P concentrations [69]. This analysis indicated that ABPs + AB−Ca reduced serum P levels, which is comparable to the results of Chen et al. [70]. ALP, a biomarker of bone formation and metabolism, is elevated when calcification is aberrant and plays a vital role in bone calcification process [16]. The ALP activity of the ABPs + AB−Ca group was substantially reduced ALP activity compared than that of the NC, CaCl_2_, and AB−Ca groups (*p* < 0.05) (Table 2). This finding is similar to that of Yang et al. [71]. Serum PINP level is a marker of bone formation [72]. PINP activity was significantly higher in the ABPs + AB−Ca group than in the NC and CaCl_2_ groups (*p* < 0.05), whereas there was no difference between the AB−Ca and ABPs + AB−Ca groups (*p* > 0.05). Serum CTX-1 is a bone turnover marker that is used to measure bone resorption. High CTX-1 levels promote bone resorption and calcium release from the bone [73]. The ABPs + AB−Ca group had substantially reduced CTX-1 activity compared to the NC and CaCl_2_ groups (*p* < 0.05), and the AB−Ca and ABPs + AB−Ca groups showed no differences (*p* > 0.05). These findings suggest that consumption of ABPs + AB−Ca can promote bone formation and prevent bone resorption in D-gal-induced aging mice.

### 3.8. Tibia Index and Ca Content Were Significantly Increased in ABPs + AB−Ca Group

Tibial properties are very sensitive to calcium absorption and metabolism and can be used as a monitoring indicator for calcium supplementation experiments [57]. Table 2 shows no substantial differences in tibial weight and length between the groups (*p* > 0.05). This finding is consistent with those of prior investigations [16]. The calcium concentration of the tibia was considerably greater in the ABPs + AB−Ca groups than in the NC, CaCl_2_, and AB−Ca groups (*p* < 0.05). Furthermore, the NC, CaCl_2_, and AB−Ca groups showed no differences in tibial calcium levels (*p* > 0.05). These findings are in line with those of previous publications [74].

Bone mineral density (BMD) is widely accepted as the gold standard for determining bone loss and osteoporosis [75]. Calcium deficiency causes SOP, which leads to a decrease in the BMD. The ABPs + AB−Ca group had considerably higher BMD than the other three groups (*p* < 0.05), while the other three groups did not differ significantly (*p* > 0.05). Compared to other peptides, casein phosphopeptides [15,16] had no discernible effect on BMD improvement. However, ABPs + AB−Ca treatment caused a 21.1% increase in BMD, which had a significant effect on the reduction of BMD caused by SOP. This finding is consistent with those of previous studies [16,76]. These studies revealed that ABPs ameliorated the bone microstructure by increasing calcium absorption.

## 4. Conclusions

In our study, the peptide–calcium chelate with a 53.68 ± 1.80% binding rate was prepared. The peptide–calcium chelate was analyzed using FTIR and SEM to characterize the binding between ABPs and AB−Ca. In vitro analysis using an intestinal Caco-2 cell model revealed that ABPs enhanced the calcium uptake activity. In vivo tests in an aging-mouse model revealed that ABPs + AB−Ca alleviated age-related bone loss and improved the bone microstructure. Collectively, our findings provide a scientific foundation for the development of novel calcium supplements and high-value use of antler bone, which may be used to avoid calcium insufficiency in the elderly.

## Figures and Tables

**Figure 1 nutrients-14-03738-f001:**
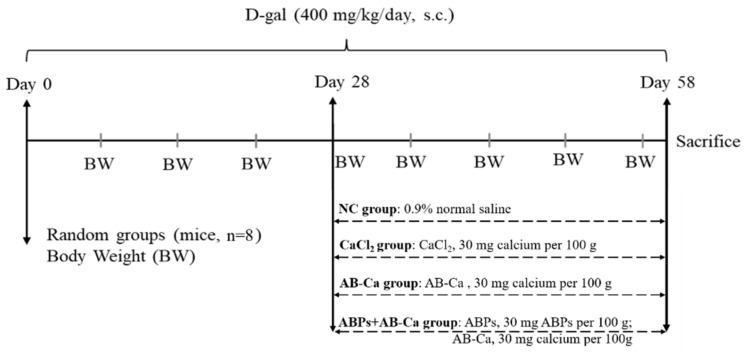
Full animal experimental protocol’s timetable. For 58 days, all groups were administered subcutaneous D-gal (400 mg/kg/day) in the morning. Each group was gavaged with 0.9% saline, CaCl_2_, AB−Ca, and ABPs + AB−Ca daily for 30 days when D-gal was injected for 28 days, and BW was recorded every week. The mice were killed on day 58.

**Figure 2 nutrients-14-03738-f002:**
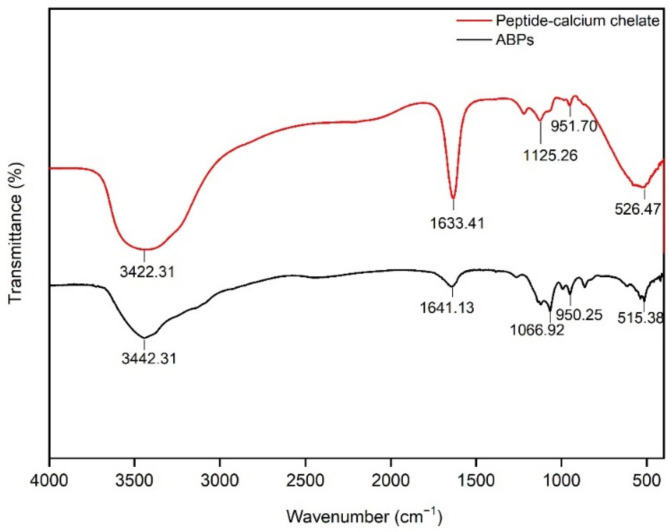
FTIR measurement of BPs and peptide–calcium chelate. ABPs (black line): antler bone peptides, peptide–calcium chelate (red line): antler bone peptide–calcium chelate.

**Figure 3 nutrients-14-03738-f003:**
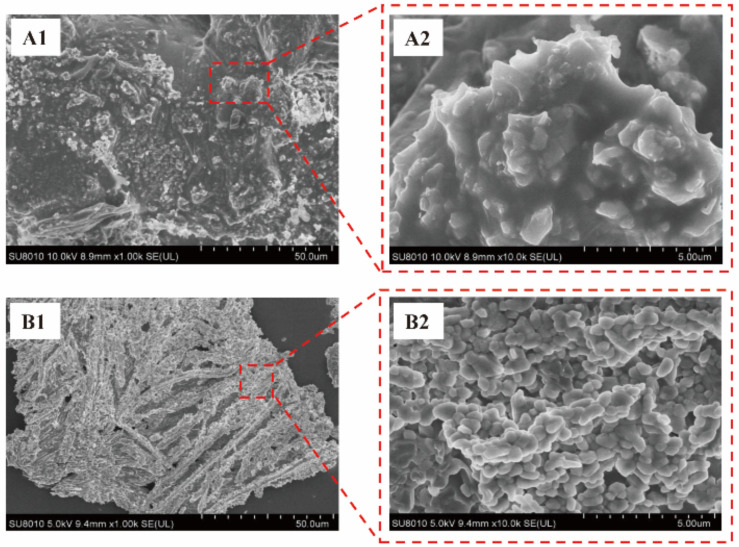
SEM pictures and structural characterization of (**A1**,**A2**) ABPs, and (**B1**,**B2**) peptide–calcium chelate. (**A1**) ABPs at ×1000 magnification; (**A2**) ABPs at ×10,000 magnification; (**B1**) peptide–calcium chelate at ×1000 magnification; (**B2**) peptide–calcium chelate at ×10,000 magnification.

**Figure 4 nutrients-14-03738-f004:**
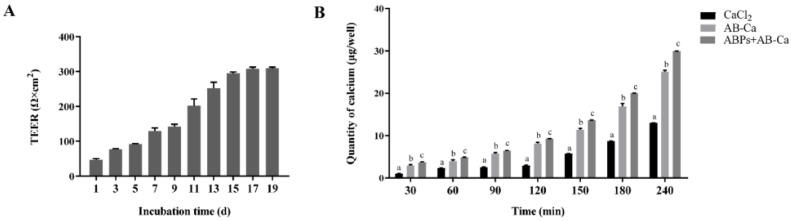
Calcium transport in intestinal Caco-2 cell was considerably enhanced by ABPs. (**A**) The TEER values of the Caco-2 cell monolayers in different days. (**B**) Calcium transport of CaCl_2_, AB−Ca, and ABPs + AB−Ca through intestinal Caco-2 cell monolayers. The columns labeled with different letters (a, b, and c) at a given point in time are significantly different (*p* < 0.05), while the same letters are not (*p* > 0.05).

**Figure 5 nutrients-14-03738-f005:**
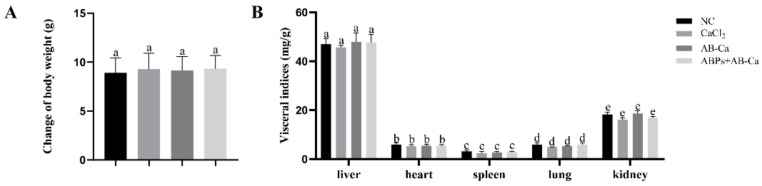
Change in body weight (**A**) and visceral index of different viscus (**B**) in different groups. Data are expressed as the mean ± SD (n = 8). Columns labeled with different letters (a, b, c, d and e ) are significantly different (*p* < 0.05), while with the same letters are not (*p* > 0.05).

**Table 1 nutrients-14-03738-t001:** ABPs were identified by their peptide sequences and molecular weight.

Peptide Sequences	Molecular Weight (Da)
Gly-Pro-Gly-Ser-Pro-Gly-Pro	567.27
Gly-Asp-Gln-Gly-Val-Pro-Gly	628.28
Gly-Pro-Ala-Gly-Pro-Gly-Pro-Pro	648.32
Gly-Ala-Pro-Gly-Pro-Ser-Gly-Pro	680.31
Gly-Pro-Gly-Pro-Ile-Gly-Asn	707.36
Gly-Ser-Pro-Pro-Ala-Thr-Ser-Cys	760.31
Gly-Pro-Ala-Gly-Pro-Pro-Gly-Ala-Pro	761.37
Gly-Glu-Pro-Gly-Lys-Gln-Gly-Pro	768.38
Gly-Ala-Pro-Gly-Pro-Ser-Pro-Gly-Pro	777.37
Gly-Ser-Pro-Pro-Gly-Glu-Gly-Ala-Pro	809.36
Gly-Ala-Pro-Gly-Pro-Pro-Ser-Gly-Gly-Pro	834.39
Gly-Glu-Pro-Gly-Pro-Glu-Gly-Pro-Ala-Gly	866.38
Gly-Glu-Arg-Gly-Glu-Gln-Gly-Ala-Pro	899.41
Gly-Pro-Pro-Gly-Glu-Pro-Gly-Pro-Pro-Gly-Pro-Pro-Gly-Pro	1208.58
Gly-Pro-Glu-Gly-Pro-Pro-Gly-Glu-Pro-Gly-Pro-Pro-Gly-Pro	1240.57
Gly-Pro-Glu-Gly-Pro-Pro-Gly-Glu-Pro-Gly-Pro-Pro-Gly-Pro-Pro	1337.63
Gly-Asp-Ile-Gly-Pro-Pro-Gly-Pro-Gln-Gly-Pro-Pro-Gly-Pro	1241.60
Gly-Glu-Val-Gly-Gln-Ile-Gly-Pro-Arg-Gly-Glu-Asp-Gly-Pro-Glu-Gly-Pro	1649.76

**Table 2 nutrients-14-03738-t002:** Serum biochemistry and tibia physicochemical indices of mice in different groups.

Measurements	NC	CaCl_2_	AB − Ca	ABPs + AB − Ca
*Serum biochemistry*				
ALP (U/L)	12.95 ± 2.92 ^a^	11.79 ± 1.83 ^a^	6.56 ± 0.84 ^b^	5.53 ± 0.40 ^b^
Ca (μmol/dL)	38.09 ± 3.74 ^a^	39.68 ± 5.16 ^a^	40.07 ± 1.67 ^a^	63.59 ± 6.12 ^b^
P (mmol/L)	3.17 ± 0.13 ^a^	3.17 ± 0.09 ^a^	3.25 ± 0.52 ^a^	2.48 ± 0.13 ^b^
PINP (ng/mL)	3.30 ± 0.56 ^a^	3.23 ± 0.19 ^a^	4.13 ± 0.05 ^b^	4.65 ± 0.33 ^b^
CTX-1 (pg/mL)	218.24 ± 15.77 ^a^	205.60 ± 10.66 ^a^	172.70 ± 6.72 ^b^	160.99 ± 1.07 ^b^
*Tibia physicochemical indices*				
Length (mm)	19.65 ± 1.20 ^a^	19.85 ± 0.55 ^a^	20.10 ± 0.11 ^a^	19.68 ± 0.49 ^a^
Dry bone weight (g)	0.045 ± 0.008 ^a^	0.048 ± 0.007 ^a^	0.050 ± 0.004 ^a^	0.045 ± 0.002 ^a^
Ca/Ash bone (μg/mg)	387.18 ± 2.57 ^a^	386.14 ± 0.10 ^a^	387.64 ± 1.71 ^a^	398.92 ± 2.55 ^b^
BMD (mg/cm^2^)	531.38 ± 34.51 ^a^	570.37 ± 2.82 ^a^	599.92 ± 34.01 ^a^	643.51 ± 57.33 ^b^

Data are presented as the mean ± SD (n ≥ 4). Mean values with different superscripts (a, b) in the same row are significantly different (*p* < 0.05), whereas the same superscripts are not (*p* > 0.05).

## Data Availability

All data generated or analyzed during this study are included in this published article. The data that support the findings of this study are available from the corresponding authors upon reasonable request.

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
