# Peer review of "Peptide−Calcium Chelate from Antler (Cervus elaphus) Bone Enhances Calcium Absorption in Intestinal Caco-2 Cells and D-gal-Induced Aging Mouse Model"

_nutrients, 2022, doi:10.3390/nu14183738_

Round 1

Reviewer 1 Report

The study entitled "Peptide-calcium chelate from antler (Cervus Elaphus) bone enhances calcium absorption in intestinal Caco‑2 cells and D-gal 2 induced aging mice model” is interesting in order to develop novel calcium supplements which might be used to avoid calcium insufficiency in the elderly.

The paper contains an interesting analysis regarding an actual and relevant topic which has biological and medical relevance. The design and analysis are appropriated, the methodology is adequate and the results are interesting. However, the study should considerate a procedure for the synthetic synthesis of peptides with therapeutic interest since obtaining directly from deer antler does not seem to be a sustainable method.

Reviewer 2 Report

The manuscript “Peptide-calcium chelate from antler (Cervus Elaphus) bone enhances calcium absorption in intestinal Caco‑2 cells and D-gal 2 induced aging mice model” present an interesting study however there are too many inconsistences in the data.

The “Materials and methods” section indicate dans the Caco-2 cells were cultured in DMEM/F-12, 10% FBS, which is not the usual protocol. Why the authors used this unusual medium?

Moreover, the protocol indicated that mice were feed with deionized water which is dangerous for health.

The condition used in this study to evaluate the calcium absorption is too far from to reality to the support a transposition to human.

To evaluate osteoblast activity determination of N terminal pro-peptides of type I procollagen (PINP) would have been more adapted.

The most important effect of ABPs+AB-Ca is an increase of calcium absorption however the consequences on the other bone parameters are not very important.

Ingestion of ABPs+AB-Ca did not have any effect on the dry bone weight and length. Moreover Ca/ash bone is only slightly increase which does not support an important increase of the BMD. Table 2 indicate that the BV/TV (%) is similar between the CaCl2 and the ABPs+AB-Ca groups however the Ca (µmol/dL) level of the CaCl2 group is similar to the NC group.  The authors did not reported an increase calcium absorption for the CaCl2 group but the data indicated an increase of  BV/TV (%) for this group that is similar to the value reported for the ABPs+AB-Ca group. Such data did not support that the ingestion of ABPs+AB-Ca despite the fact than an increase calcium absorption was measured have any interest to improve bone quality. Usually, a good correlation is reported between the BMD and the BV/TV (%) which is not reported in the present study.

English editing is needed.

Round 2

Reviewer 2 Report

The manuscript “Peptide-calcium chelate from antler (Cervus Elaphus) bone enhances calcium absorption in intestinal Caco‑2 cells and D-gal 2 induced aging mice model” present an interesting study that can be published.

However the impact of that study is low.

 As previously stated the condition used in this study to evaluate the calcium absorption is too far from to reality to the support a transposition to human.

Moreover the authors did not reported significant differences for the CTX and PINP serum level between the AB-Ca and ABPs+AB-Ca which indicate that the osteoblasts and osteoclasts activities are similar in both group. Mores studies are needed to explain the mechanisms responsible of the BMD increase by only an increase of the calcium serum.